# Differential Action of Silver Nanoparticles on ABCB1 (MDR1) and ABCC1 (MRP1) Activity in Mammalian Cell Lines

**DOI:** 10.3390/ma14123383

**Published:** 2021-06-18

**Authors:** Damian Krzyzanowski, Marcin Kruszewski, Agnieszka Grzelak

**Affiliations:** 1Department of Pediatrics, Oncology and Hematology, Medical University of Lodz, 91-738 Lodz, Poland; 2Laboratory of Epigenetics, Institute of Medical Biology, Polish Academy of Sciences, 93-232 Lodz, Poland; 3Centre for Radiobiology and Biological Dosimetry, Institute of Nuclear Chemistry and Technology, 03-195 Warsaw, Poland; m.kruszewski@ichtj.waw.pl; 4Department of Molecular Biology and Translational Research, Institute of Rural Health, 20-090 Lublin, Poland; 5Department of Molecular Biophysics, University of Lodz, 90-237 Lodz, Poland; agnieszka.grzelak@biol.uni.lodz.pl

**Keywords:** silver nanoparticles, ABC transporters, ATP-binding cassette, multidrug resistance, P-glycoprotein

## Abstract

Silver nanoparticles (AgNPs), due to their unique properties have been receiving immense attention in recent years. In addition to their antibacterial and antifungal activities, AgNPs also cause apoptosis, mitochondria disfunction, nucleic acid damage and show potent anticancer properties in both multidrug resistance (MDR) and sensitive tumors. The MDR phenomenon, caused by the presence of ATP-binding cassette (ABC) proteins, is responsible for the failure of chemotherapy. Thus, investigating the influence of widely used AgNPs on ABC transporters is crucial. In the present study, we have examined the cytotoxicity of silver nanoparticles of a nominal size of 20 nm (Ag20) on the cell lines of different tissue origins. In addition, we have checked the ATP-binding cassette transporters’ activity and expression under AgNP exposure. The results indicate that Ag20 shows a toxic effect on tested cells, as well as modulating the expression and transport activity of ABC proteins.

## 1. Introduction

Nanotechnology is a relatively new field of science connecting the achievements of biology, chemistry, physics and engineering. Synthetic nanomaterials, defined as any molecules of a diameter smaller than 100 nm, have received tremendous attention and have found widespread applications in life sciences as well as everyday life, due to their unique designs and property combinations, compared with bulk materials. Their small size and developed surface, resulting in a large specific surface to volume ratio, make them highly reactive molecules with unique physical and chemical properties [1,2].

Among the list of thousands of commercially available products, nanomaterials containing silver nanoparticles (AgNPs) are mostly used. Nanosilver has effective antibacterial, antiviral and antifungal properties. However, the mechanism of action of AgNPs is still not well understood. The development of nanotechnology and the extensive use of AgNPs are provoking discussion about the toxicity of these molecules on the environment in the direct way, or through indirect action [3,4]. Considering such a wide spectrum of applications for AgNPs, one must be wary of exposure to them [5]. AgNPs penetrate the human organism through the respiratory and digestive systems, as well as through the skin, both intact and damaged [6]. They also penetrate the blood—brain or blood—testis barriers. Because of their small size, AgNPs can be easily transported through the blood circulation system [7]. The transport of AgNPs inside cells begins with their internalization and translocation into lysosomes. The main mechanisms of AgNPs penetrating into the cell are via clathrin-dependent endocytosis [8], mediated by caveolae (i.e., clathrin-independent endocytosis), lipid-dependent endocytosis and, to a lesser extent, micropinocytosis [9], but also through direct diffusion through the cell membrane into the cytoplasm [10,11,12]. During endocytosis, nanoparticles are trapped in the early endosome formed from the cell membrane, which then transforms into the late endosome and finally the lysosome. The penetration of AgNPs is a process strictly dependent on the incubation time, concentration, size, shape, surface and charge modification [13]. Smaller nanoparticles (>20 nm) are biologically more effective (they can penetrate the cell nucleus or mitochondria [14]).

When AgNPs are internalized, they induce a number of biological effects, including: intracellular oxidative stress by generation of reactive oxygen species and, in consequence, stimulation of the IκB kinase/nuclear factor-κB (IKK/NF-κB) and nuclear factor-erythroid 2-related factor 2/Kelch-like ECH-associated protein 1 (Nrf2/KEAP1) pathways [15,16], and cell membrane impairment and damage. As an indirect action, they trigger the programmable cell death—apoptosis, inflammatory processes and disorders of mitochondrial functioning [17,18]. They exert a significant inhibitory effect on RNA by suppressing the transcription through RNA polymerase-silver binding [19] and cause DNA damage, genotoxicity, chromosomal aberrations, inhibition of proliferation and many others [1,20,21,22].

Many studies in the field of AgNP toxicity have been carried out, giving the evidence of the remarkable anticancer features. However, there is little information concerning the effect of these molecules on the ABC transporters, which are particularly important from a clinical point of view. There are data suggesting that quantum dots are actively co-transported with cholesterol by ABCB1 [23]. On the other hand, nanoparticles have been proposed as vehicles allowing the introduction of bound or encapsulated drugs to circumvent the activity of multidrug transporters. Although there is evidence that AgNPs have a notable antiproliferative effect, induce apoptosis mediated cell death, cause cell cycle arrest, and modulate ABC transporter activity in both drug sensitive and in the multidrug resistant cancer cells [24], there is a little information on the effect of AgNPs on the MDR protein expression. Thus, further investigation in this area is strongly required.

ATP-binding cassette (ABC) transporters are present in all living organisms, from bacteria to man, and they are among the largest and most widely expressed protein superfamilies known [25]. In humans, 49 ABC genes have been described and organized into seven subfamilies (A−G) named according to the nomenclature based mainly on a sequence homology [26,27]. The ABC superfamily consists mainly of membrane proteins that transport various types of substrates, including sugars, amino acids, lipids, sterols, peptides, endogenous metabolites, ions and wide spectrum of xenobiotics [28,29]. A common feature of the above-mentioned proteins is the fact that they are active transporters—they pump substrates against the concentration gradient using the energy from ATP hydrolysis. ABC transporters are distinguished by highly conservative amino acid sequences (transmembrane domains TMDs) located in the nucleotide-binding domain (NBD), so-called Walker A and Walker B motifs, separated by an “ABC signature” motif responsible for transport [30,31]. The overexpression of ABC proteins is the main reason for the MDR phenomenon and may potentially be involved in conferring resistance to chemotherapeutic agents as well as leading to anticancer therapy failure.

From a clinical point of view, the key role in the above-mentioned mechanism is played by three ABC proteins: P-glycoprotein (ABCB1/MDR1) [32], multidrug resistance protein 1 (ABCC1/MRP1) [33] and breast cancer resistance protein (ABCG2/BCRP) [34].

The aim of this study was to examine in vitro the effect of AgNPs on the regulation of ABC transporter expression as well as the activity in the cells of different tissue origins, selected on the basis of in vivo deposition in the animal models [35,36].

## 2. Materials and Methods

### 2.1. Reagents

The highest-purity reagents available were used to perform all experiments. Dulbecco’s Modified Eagle Medium (DMEM), phosphate-buffered saline (PBS), fetal bovine serum (FBS) and trypsin were from Gibco (Thermo Fisher Scientific, Waltham, MA, USA). Bare silver nanoparticles of nominal size of 20 nm (Ag20) were purchased from Plasmachem GmbH (Berlin, Germany). MycoProbe Mycoplasma Detection Kit were from R&D Systems (Minneapolis, MN, USA). All other reagents were purchased from Sigma-Aldrich (St. Louis, MO, USA) or Thermo Fisher Scientific (Waltham, MA, USA) unless stated otherwise.

### 2.2. Cell Culture

Non-small cell lung carcinoma: A549 (CCL-185), hepatocellular carcinoma: HepG2 (HB-806), and colorectal adenocarcinoma: SW620 (CCL-22) were purchased from the American Type Culture Collection (ATCC, Manassas, VA, USA) and cultured in 75 cm^2^ cell culture flasks (Nunc, Roskilde, Denmark) in Dulbecco’s Modified Eagle Medium buffered with HEPES to pH range of 7.1–7.3, supplemented with Glutamax-I and 10% (*v*/*v*) fetal bovine serum (Thermo Fisher Scientific, Waltham, MA, USA) at standard conditions: 37 °C in a humidified-air atmosphere containing 5% carbon dioxide. Passages were carried out when the cultures reached 70–85% confluence. The cells were split with 0.25% trypsin-EDTA, counted by the Countess Automated Cell Counter (Life Technologies, Waltham, MA, USA), and plated at 4 × 10^4^ cells/cm^2^ in a new flask or used for experiments. All cells were free of Mycoplasma (tested with MycoProbe Mycoplasma Detection Kit by R&D, Minneapolis, MN, USA) and were harvested in the exponential growth phase before use.

### 2.3. Silver Nanoparticle Preparation

A stock solution (2 mg/cm^3^) of bare AgNPs with nominal diameters of 20 nm (Ag20) was prepared freshly before each experiment. Briefly, Ag20 was suspended in 800 mm^3^ of Mili-Q water, mixed carefully and sonicated with OmniRuptor 4000 with 4.2 kJ/cm^3^ total ultrasound energy (Omni International, Kennesaw, GA, USA). Then 100 mm^3^ of 15% bovine serum albumin and 100 mm^3^ of 10-fold concentrated phosphate buffered saline (PBS) were given immediately to aliquots of suspension, mixed well and used to carry out the experiments. The nanoparticles and their behavior in the culture media have been described previously [37,38].

### 2.4. Cytotoxicity Assay

Cytotoxicity of Ag20 was determined using the neutral red assay which is based on the ability of viable cells to incorporate and bind the supravital dye neutral red in the lysosomes (reductants-independent assay). Briefly, 15 × 10^3^ SW620, A549 and HepG2 cells per well in 100 mm^3^ medium were seeded in 96-well tissue culture plates (Nunc, Roskilde, Denmark) and after at least 24 h were treated for the appropriate period of 24, 48 or 72 h with Ag20 in the concentration range of 0–300 µg/cm^3^. Then 20 mm^3^ of 0.33% (*w*/*v*) neutral red solution in PBS was added and incubated for an additional 2 h at standard conditions. The cells were subsequently washed with PBS and dye was extracted in each well using neutral red fixer: 50% (*v*/*v*) ethanol with 1% (*v*/*v*) acetic acid in Mili-Q water (Merc Millipore (Billerica, MA, USA). Then plates were shaken 10 min on Thermomixer Comfort (Eppendorf, Hamburg, Germany) and the absorbance was measured using spectrophotometer EnVision Multilabel Reader (Perkin Elmer, Waltham, MA, USA) at 540 nm.

### 2.5. RNA Isolation and Real-Time PCR Gene Expression Analysis

SW620, A549 and HepG2 cells were seeded in 6-well plate (Nunc, Roskilde, Denmark) at density 5 × 10^5^ per well. After 24 h of culture, cells were treated with 50 µg/cm^3^ Ag20 for 1, 2, 4, 24 h. Subsequently, cells were rinsed with PBS, trypsinized and total RNA was isolated, employing MagNA Pure LC 2.0 Instrument (Roche, Basel, Switzerland) according to the manufacturer’s protocol. After genomic DNA removal by RNase free, DNase I digestion, total RNA of 1 µg was reverse-transcribed using the SuperScript III First-Strand Synthesis SuperMix (Life Technologies, Waltham, MA, USA). PCR analysis was performed using C1000 Thermal Cycler—CFX384 Real-Time System (Bio-Rad, Hercules, CA, USA) and RealTime ready Human ABC Transporter Panel (Roche, Basel, Switzerland). DNA was omitted in non-template control. The amount of target mRNA in the various samples was estimated using the 2^−ΔCT^ relative quantification method employing Rest2009 v2.0.13 (Qiagen, Hilden, Germany) and presented as heatmaps [39]. Though the same gene panels were used for all cell lines, in some cases expression of a particular gene was under detection limit. In this case the gene is omitted on a heatmap for this cell line.

### 2.6. Western Immunoblotting

The cellular proteins were extracted from SW620, A549 and HepG2 cells with RIPA buffer containing Halt Protease Inhibitor Cocktail (Thermo Fisher Scientific, Waltham, MA, USA), and equal protein amounts (protein concentration of lysates measured by BCA assay, Thermo Fisher Scientific, Waltham, MA, USA) were subjected to SDS-PAGE. After electrophoresis proteins were transferred onto Trans Blot Turbo Mini PVDF membranes with Trans-Blot Turbo Transfer System (Bio-Rad, Hercules, CA, USA), blocked with 5% BSA and incubated with specific primary antibodies: mouse anti-human ABCB1 (clone F4, Sigma Aldrich, St. Louis, MO, USA). Specific HRP-conjugated secondary antibodies were used (Sigma Aldrich, St. Louis, MO, USA), and protein bands were detected using Pierce ECL Western Blotting Substrate (Thermo Fisher Scientific, Waltham, MA, USA) employing Mini HD (Uvitec, Cambridge, UK). MDR1/ABCB1 Sf9 insect membranes (Solvo Biotechnology, Budaörs, Hungary) were used as a positive control for the presence of ABC proteins, and β-actin serves as an internal control.

### 2.7. Flow Cytometry

#### 2.7.1. Calcein Accumulation

To investigate ABCB1 activity, the calcein assay was performed according to Richter et al. [40] with minor modifications. Calcein acetoxymethyl ester (Cal AM) solution was added to the cells suspension to a final concentration of 200 nM and the kinetic intensity of the median fluorescence was immediately recorded using an LSRII flow cytometer every 2 min for 12 min at 37 °C on the emission channel 530/20 nm to ensure initial conditions. The fluorescence increased with increasing inhibition of ABCB1-mediated efflux of calcein AM. The ABCB1 activity was determined by calculating the slope of the fluorescence recorded over 12 min, plotted relative to the fluorescence in the presence or absence of 10 μM verapamil. The calcein assay was analyzed using the slope of the linear curve determined without inhibitor subtracted from the slope of the linear curve in the presence of the inhibitor. Verapamil-dependent calcein accumulation was calculated.

#### 2.7.2. BCECF Efflux

To investigate ABCC1 activity the BCECF efflux assay was performed according to Bachmeie et al. [41] with minor modification. (2′,7′-Bis-(2-Carboxyethyl)-5-(and-6)-Carboxyfluorescein (BCECF AM) solution was added to the cell suspensions to a final concentration of 1 µM and incubated 10 min at 37 °C. Then the cells were washed with ice cold PBS, resuspended in fresh medium and split into two samples: without inhibitor and with 10 μM MK571. The kinetic intensity of the fluorescence median was recorded using an LSRII flow cytometer every 15 min for 120 min at 37 °C on the emission channel 530/20 nm. The BCECF assay was analyzed using the area under curve (AUC) of plotting a time-dependent median fluorescence of BCECF curve in the presence or absence of MK571. MK571 dependent BCECF efflux was calculated by subtracting AUC without inhibitor from AUC with inhibitor.

### 2.8. Statistical Analysis

Data are expressed as an arithmetic mean ± standard deviation (SD), standard error of the mean (SEM) or confidence interval (for curve fitting and IC_50_ calculations) from at least three independent replicates. In order to show statistically significant differences, the student’s *t*-test and one- or two-way ANOVA with Dunnett’s or Bonferroni’s post hoc test were used. All statistical analyses were performed using GraphPad Prism 7.05 (Sandiego, CA, USA). The difference between the mean experimental values was considered statistically significant for *p* ≤ 0.05.

## 3. Results

In this study we examined the effect of previously defined Ag nanoparticles [37] with a nominal size of 20 nm on three cell lines of different tissue origin: non-small-cell lung carcinoma (A549), hepatocellular carcinoma (HepG2) and colon carcinoma (SW620).

### 3.1. Cell Viability Measurement

To compare the cytotoxicity effect of Ag20 on A549, HepG2 and SW620, cells were treated with Ag20 with a concentration range from 0–300 µg/cm^3^ and their viability was determined after 24, 48 and 72 h with neutral red assay. The survival curves are shown in Figure 1, and the IC_50_ values are presented in Table 1. The analysis of the IC_50_ parameter showed that the A549 cell line was the most resistant to the effects of AgNPs, where it was only possible to determine IC_50_ value after 72 h of incubation, and was 202.7 μg/cm^3^. The HepG2 cell line was the most sensitive, with the IC_50_ for 24, 48 and 72 h respectively: 26.1; 21.5 and 15.8 μg/cm^3^. As for the SW620 colorectal cancer cell line, the IC_50_ was for 24, 48 and 72 h: 59.0; 53.8 and 39.4 μg/cm^3^, respectively.

### 3.2. Gene Expression Analysis

After cytotoxicity determination, the effect of Ag20 on the expression profile of the genes encoding the human ABC transporter superfamily in A549, HepG2 and SW620 cell lines was checked using the qPCR method. Figure 2 shows the expression profile of the ABC genes presented as heatmaps. The results show that in A549 and HepG2 cells (Figure 2A,B) after short incubation times with 50 μg/cm^3^, AgNPs caused an increase in the expression of ABC transporters except for a few cases, and after 12 h of incubation, a significant decrease. Interestingly, in the case of SW620 cell line, an opposite trend was observed. (Figure 2C).

### 3.3. Transport Activity

#### 3.3.1. Calcein Accumulation

Extracellularly nonfluorescent, noncharged calcein acetoxymethyl ester penetrates freely inside the cell, where it undergoes hydrolysis, resulting in the formation of a fluorescent calcein. Cal AM is a substrate for ABCB1. The use of a relatively specific ABCB1 inhibitor—verapamil allows the assessment of the activity of the ABCB1 protein by determining the contribution of this protein to Cal AM transport. Examples of the plot of the kinetics of the calcein accumulation are shown in Figure 3A–C. Based on the calcein accumulation plot in A549 cells treated with Ag20, there were no significant differences in the ABCB1 activity compared to control cells within the tested concentrations (Figure 3D). For the HepG2 and SW620 cell lines, a significant decrease in the accumulation of the fluorescent substrate was noted after 2 h incubation with 50 μg/cm^3^ of AgNPs (Figure 3E,F).

#### 3.3.2. BCECF Efflux

Likewise calcein AM, BCECF in the form of acetoxymethyl ester penetrates into the cells, where it undergoes hydrolysis to a free fluorescent dye in anionic form. The anion is a substrate of the ABCC1 transporter, thus the ABCC1 activity assay is based on the measurement of the intracellular fluorescence decay. An example of the plot of the BCECF efflux kinetics is shown in Figure 4A–C. The results show that the AgNPs did not have a significant effect on the inhibitor-dependent BCECF efflux in any tested cells, which might prove the lack of influence on the activity of the ABCC protein (Figure 4E,F). Moreover, for A549 cells, low values of MK571-dependent BCECF efflux may indicate extremely low activity of the above proteins in this line (Figure 4D).

### 3.4. Western Immunoblotting

The densitometric analysis of the expression of ABCB1 in A549, HepG2 and SW620 cells was performed under the influence of 25 and 50 μg/cm^3^ of Ag20 to check if the decrease in P-glycoprotein activity was not associated with decreased protein expression (Table 2). There was a slight decreasing tendency in the expression of protein in A549 and SW620 (Figure 5A,C) cells treated with Ag20 and no differences in HepG2 cells compared to control cells (Figure 5B).

## 4. Discussion

Growing interest in nanomaterials and their increasing use make humans vulnerable to exposure to them, and therefore there is an urgent need to address nanoparticle risk assessment. Based on the in vivo studies, it has been shown that AgNPs enter the circulatory system both by inhalation and after ingestion, and then are further deposited in many organs, mainly the lungs, liver, spleen, kidneys, brain, heart, testes and intestines [42,43]. Thus, for experimental purposes, we chose SW620, A549 and HepG2 cell lines corresponding to the colon, lung and liver tissue, respectively.

Analysis of AgNP cytotoxicity showed that all tested cell lines responded in a concentration dependent manner. However, the A549 cell line was the most resistant, whereas the HepG2 cell line was the most sensitive. These results are in line with previous data [44], where the proposed explanation of this phenomenon was the fact that A549 cells are more resistant to stress inducing factors, such as xenobiotics, due to the mutations in the main pathways activated in response to oxidative stress—nuclear factor-erythroid 2-related factor 2/Kelch-like ECH-associated protein 1 (NRF2/KEAP1) [45], nuclear transcription factor κB (NF-κB) and K-*Ras* [46,47]. This is in line with the general picture of AgNP toxicity. Silver nanoparticles easily penetrate into the cells and release water-soluble toxic ionic silver Ag^+^ within the cytoplasm, which is very reactive [48]. Moreover, AgNPs and released Ag^+^ silver ions strongly react with thiol groups of reduced glutathione and other proteins, such as thioredoxin or thioredoxin peroxidase that may affect their activity and further potentiate oxidative stress and cellular toxicity. Kovács et al. concluded that the reduced viability of Colo205 and Colo320 (P-glycoprotein overexpressing cells) caused by AgNPs exposition was mediated by caspase 3 dependent apoptosis and attenuation of oxidative stress messengers, such as: *p21*, *survivin* and *sod-1* genes [24].

The main aim of the study was to investigate the effect of AgNPs on the modulation of expression and activity of the MDR proteins in cancer cells of various tissue origin. Since the ability of AgNPs to induce oxidative stress is well documented [49,50,51,52], they should have also the potential to regulate many physiological processes, dependent on cellular redox status. Among others, expression of some MDR proteins might be regulated by oxidative stress dependent factors, such as NF-κB that is one of the key transcription factors activated by reactive oxygen species. NF-κB might be involved in the regulation of MDR protein expression as a promoter region domain of the *MDR1* gene contains NF-κB binding sites [53]. Indeed, Terada et al. showed that P-glycoprotein expression in Caco-2 cells was increased in response to 1 µM of hydrogen peroxide, but decreased upon exposure to the compound at a concentration of 10 mM [54].

In this work, the analysis of ABC gene expression has also shown that in A549 and HepG2 cells, a short time of incubation with 50 μg/cm^3^ Ag20 resulted in an increase of the ABC proteins’ mRNA level during the 6 h after treatment, however a clear decrease was observed after 12 h. The opposite situation was noticed in the case of SW620 cells, where an initial decrease (up 2 h) in the mRNA level of ABC proteins was accompanied by its gradual increase (Figure 2). The most likely reason for these differences may be the disparate expression levels of ABC transporters in the tested lines. Szakács et al. have profiled mRNA expression of the 48 known human ABC transporters in 60 diverse cancer cell lines used by the National Cancer Institute to screen for anticancer activity. Results clearly indicate differences in the expression of ABC proteins in A549 and SW620 cell lines [55].

In the further work, we focused on closer characterization of the functional activity of the ABCC1 and ABCB1 proteins, which are important from the clinical point of view. Analysis of the functional activity of ABCC1 and ABCB1 proteins revealed a reduced verapamil-dependent calcein accumulation in HepG2 and SW620 cells, reflecting the reduced activity of ABCB1, and no difference in A549 cells (Figure 3). In line with this, the expression of the ABCB1 protein was decreased for almost all the times and AgNPs concentrations tested (Figure 5). These results are in concordance with literature data. Kovács et al. showed in a colorectal cancer model that biocompatible AgNPs coated with citrate, about 28 nm in diameter, modulated the expression and activity of P-glycoprotein in Colo320 cells. The authors concluded that the decreased expression of the transporter contributed to the decreased activity of this protein, limiting the efflux of rhodamine 123—an ABCB1 substrate. However, the question still remains whether AgNPs exert a direct inhibitory effect on the ABC transporter or just disrupt mitochondrial function and impaired ATP production, or perhaps through transcriptional silencing of the *MDR1* gene [24].

An inhibited efflux of calcein was also observed in the MDCKII-MDR1 cell line treated with AgNO_3_ and AgNPs (23 nm). Interestingly, only AgNPs increased the calcein accumulation in wild type MDCKII cell line, that might suggest the direct interaction of NPs with proteins, rather than the indirect action through thiol group inhibition or mitochondrial interactions. The authors confirmed the observed phenomenon in an in vivo model using *Daphnia magna*, where an increased accumulation of calcein was observed after 60 min exposure to AgNPs [56]. The inhibition of ABCB1 by citrate-modified AgNP function was also observed in MCF-7 and MCF-7/KCR (cells with ABCB1 overexpression). While the treatment of MCF-7/KCR cells with 5 nm AgNPs had no effect on the transport activity of the ABCB1 protein, treatment with a 75 nm particle accumulation of rhodamine 123 inhibited the P-glycoprotein. Interestingly, the authors did not observe statistically significant changes in the amount of P-glycoprotein in MCF-7/KCR cells after 65 h incubation with both 5 and 75 nm AgNPs. They concluded that reduced ABCB1 transporter activity was not coupled to modulated protein expression in MCF-7/KCR cells, but rather to disturbances in mitochondrial activity, misfolding of ABCB1 protein causing ER stress and mistrafficking [57]. These data are partially in line with our results. We have noticed a similar phenomenon of decreased activity of ABCB1 after 2 h incubation with Ag20, without affecting protein expression in HepG2 and SW620 cells. Interestingly, no such effect was observed for longer incubation times.

On the contrary, despite the fluctuations of ABCC proteins’ mRNA levels, no effect of AgNPs on the functional activity of transporters from the ABCC subfamily was observed on any tested cell lines and incubation times (Figure 2). This suggests different mechanisms of regulation of expression and activity of ABCC1 and ABCB1 proteins.

Discordant mRNA levels and ABCB1 protein expression due to AgNP treatment are not fully understood. The regulation of gene product expression involves a series of linked processes that contribute to establishing the rates of protein production and turnover. These include, among others: translation rates (influenced by the mRNA sequence—e.g., upstream open reading frames); translation rate modulation (modulated through the binding of proteins or noncoding RNA to regulatory elements on the transcript); modulation of a protein’s lifespan by the complex ubiquitin−proteasome pathway; or protein synthesis delay (transcript changes affect protein levels with a particular delay) [58,59,60,61]. All these processes might be regulated under oxidative stress [62,63].

Concluding, the obtained results show the complexity of the mechanism of action of AgNPs on transcriptional regulation, post-translational processing and the activity of proteins related to the phenomenon of MDR. It is not known yet whether the AgNPs act directly on the transporter by binding to the protein, thus changing its activity, or indirectly by blocking the ATPase activity or inducing oxidative stress and modulating ABC protein expression. Further research is necessary to elucidate the exact mechanism of AgNP action on ABC proteins’ functionality.

## Figures and Tables

**Figure 1 materials-14-03383-f001:**
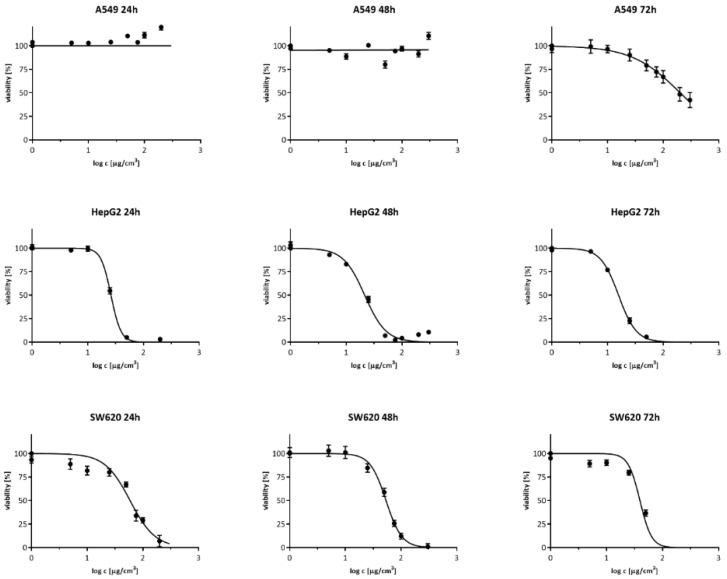
Viability of A549, HepG2 and SW620 as a function of the AgNP concentration; arithmetic mean ± SEM; *n* = 3, neutral red test. Incubation time: 24, 48 and 72 h.

**Figure 2 materials-14-03383-f002:**
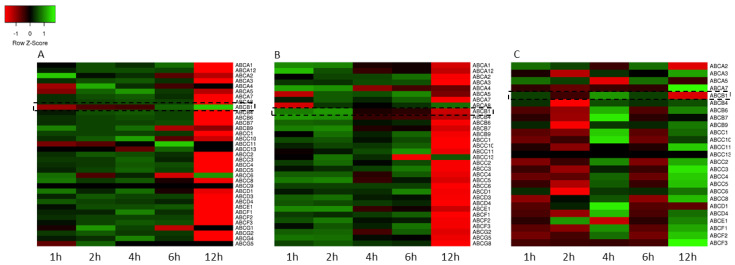
Heatmaps representing the expression profile of the genes encoding ABC transporters in A549 (**A**), HepG2 (**B**) and SW620 (**C**) cell lines treated for 1, 2, 4, 6 and 12 h with 50 μg/cm^3^ Ag20. Rows represent genes and columns represent incubation time. The intensity of each color represents the standardized ratio between each value and the average expression of each gene in all samples. Data expressed as the arithmetic mean of relative gene expression normalized to untreated cells; *n* = 3.

**Figure 3 materials-14-03383-f003:**
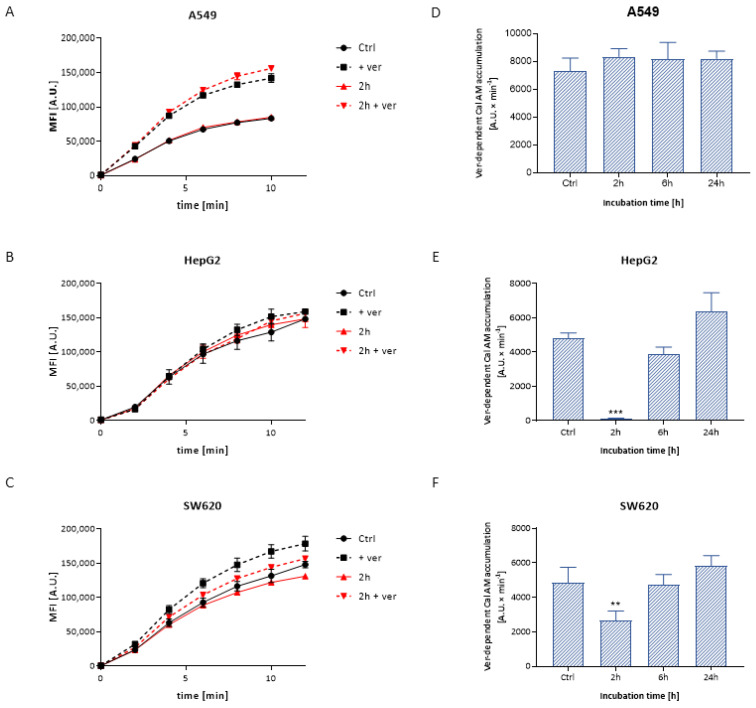
Illustrative plot of intracellular Cal AM accumulation versus time in A549 (**A**), HepG2 (**B**) and SW620 (**C**) cells treated with Ag20 in the presence and absence of verapamil. Verapamil-dependent calcein AM accumulation in A549 (**D**), HepG2 (**E**) and SW620 (**F**) cells treated for 2, 6 and 24 h with 50 μg/cm^3^ Ag20 expressed as the slope of the rectilinear segment of the fluorescence increment over time. One-way ANOVA and Dunnett’s post hoc test were used. Significance level: ** *p* < 0.01; *** *p* < 0.001. Data expressed as arithmetic mean ± SD; *n* = 3.

**Figure 4 materials-14-03383-f004:**
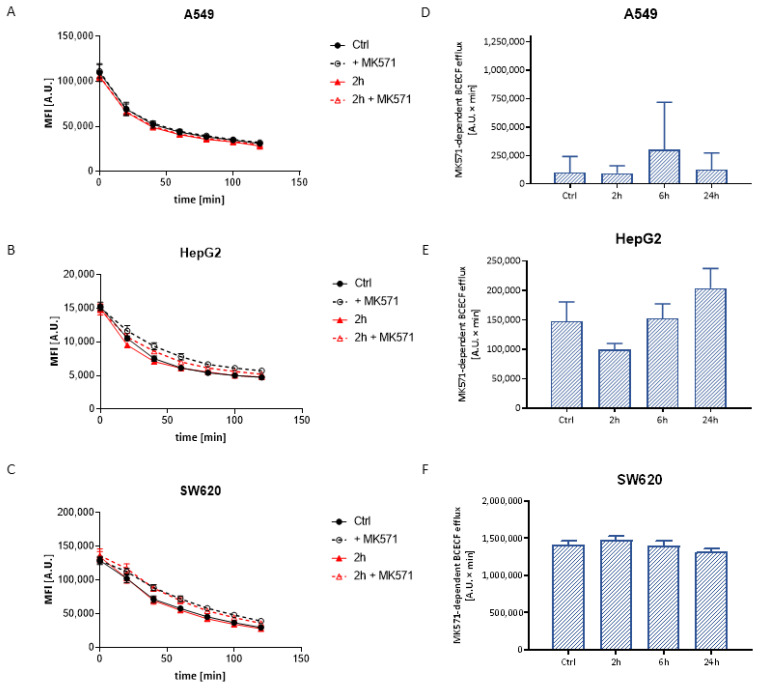
Illustrative plot of intracellular BCECF efflux versus time in A549 (**A**), HepG2 (**B**) and SW620 (**C**) cells treated with Ag20 in the presence and absence of MK571. MK571-dependent BCECF efflux in A549 (**D**), HepG2 (**E**) and SW620 (**F**) cell lines treated for 2, 6 and 24 h with 50 μg/cm^3^ Ag20 expressed as the area under the curve of the fluorescence decay over time. One-way ANOVA and Dunnett’s post hoc test were used. Data expressed as arithmetic mean ± SD; *n* = 3.

**Figure 5 materials-14-03383-f005:**
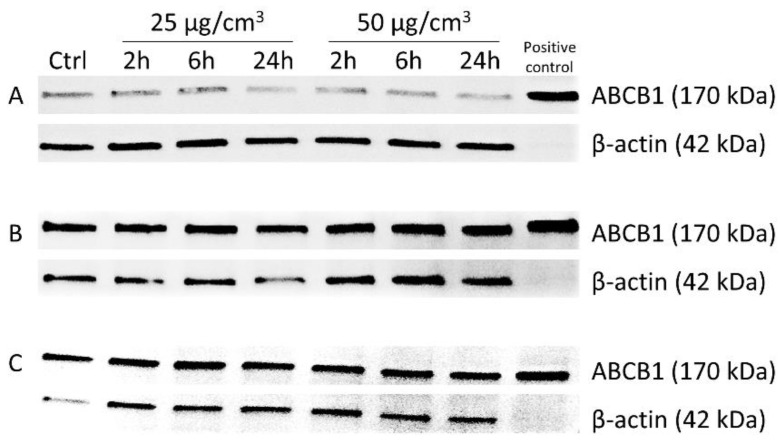
Immunodetection of ABCB1 in A549 (**A**), HepG2 (**B**) and SW620 (**C**) cells treated with 25 and 50 μg/cm^3^ Ag20 for 2, 6 and 24 h. MDR1/ABCB1 Sf9 insect membrane was used as a positive control for the presence of ABC protein, and β-actin serves as an internal control.

**Table 1 materials-14-03383-t001:** The summary of Ag20 cytotoxicity in A549, HepG2 and SW620 cell lines. The results are presented as the IC_50_ mean value (µg/cm^3^) ± the 95% confidence interval, *n* = 3. “—” was not determined for the IC_50_ parameter.

Time	24 h	48 h	72 h
A549	—	—	202.7
	—	—	(184.9–222.1)
HepG2	26.1	21.5	15.8
	(25.3–27.0)	(20.0–23.1)	(14.9–16.8)
SW620	59.0	53.8	39.4
	(51.9–67.0)	(49.5–58.6)	(34.4–45.1)

**Table 2 materials-14-03383-t002:** Densitometry quantification of ABCB1 protein expression. Data normalized to untreated cells.

Dose	25 μg/cm^3^			50 μg/cm^3^		
Time	2 h	6 h	24 h	2 h	6 h	24 h
A549	0.81	0.87	0.77	0.83	0.75	0.70
HepG2	1.02	1.03	1.15	1.06	0.91	0.90
SW620	0.70	0.72	0.64	0.59	0.66	0.69

## Data Availability

The data presented in this study are available on request from the corresponding author.

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
