# Peer review of "Differential Action of Silver Nanoparticles on ABCB1 (MDR1) and ABCC1 (MRP1) Activity in Mammalian Cell Lines"

_materials, 2021, doi:10.3390/ma14123383_

Round 1
Reviewer 1 Report
Executive Summary
The manuscript titled “Differential action of silver nanoparticles on ABCB1 (MDR1) and ABCC1 (MRP1) activity in mammalian cell lines” investigated the potential usage of Silver Nanoparticles with 20 nm size distribution in cancer-related research. Overall, the manuscript is scientifically written. The authors may perform minor revisions before the final publication.
Major Comments
- Please work with the editorial office to proofread the whole manuscript. The authors need to improve their skills in scientific writing.
- Oxidative stress may not directly be correlated to Ag20. I do not think it is necessary to discuss that as part of this research unless supported by experimental data.
Minor Comments
- Figure 4, please update the footnote to explain the meaning of Figures 4-D and 4-E, which is consistent with Figure 3. Further, please incorporate data for A549 cells.
Reviewer 2 Report
In this manuscript by Krzyzanowski et al., the authors discussed the influence of silver nanoparticles of 20 nm on the expression and activity of ABC proteins using different cell lines (SW620, A549 and HepG2). The data shown in the manuscript are solid, and the results are interesting to researchers who work on the toxicity of nanomaterials. This paper should be accepted after the following minor issues addressed:
- More information of the Ag nanoparticles should be provided, such as the size distribution, and surface functionalization. These properties may also influence the results.
- In Figure 1, the viability curve of A549 at 24 hours seems to show an increased viability with the increase of concentration, and for many data points, the data are >100%. The authors should make sure this is accurate.
- The authors included a long discussion on reactive oxygen species in the Discussion section. Experiments should be performed to directly measure the ROS. Otherwise, the discussion on ROS can be shortened.
Reviewer 3 Report
The work is well conceived and presented. I would accept the manuscript in the present form.
